# Comparison of the Effectiveness of Two Different Vaccination Regimes for Avian Influenza H9N2 in Broiler Chicken

**DOI:** 10.3390/ani10101875

**Published:** 2020-10-14

**Authors:** Shaimaa Talat, Reham R. Abouelmaatti, Rafa Almeer, Mohamed M. Abdel-Daim, Wael K. Elfeil

**Affiliations:** 1Department of Birds and Rabbits Medicine, Faculty of Veterinary Medicine, Sadat City University, Menoufiya 32958, Egypt; shaimaa610@gmail.com; 2Department of Animal Epidemiology and Zoonosis, Sharkia Veterinary Directorate, General Organization of Veterinary Services (GOVS), Ministry of Agriculture, Sharkia 44511, Egypt; r.abouelmaatti@yahoo.com; 3Department of Zoology, College of Science, King Saud University, Riyadh 11451, Saudi Arabia; ralmeer@ksu.edu.sa (R.A.); abdeldaim.m@vet.suez.edu.eg (M.M.A.-D.); 4Pharmacology Department, Faculty of Veterinary Medicine, Suez Canal University, Ismailia 41522, Egypt; 5Avian and Rabbit Medicine Department, Faculty of Veterinary Medicine, Suez Canal University, Ismailia 41522, Egypt

**Keywords:** avian influenza, homologous vaccine, heterologous vaccine, broiler, early infection

## Abstract

**Simple Summary:**

The low pathogenic avian influenza H9N2 virus has been associated with severe economic losses in broiler chicken flocks. Problems associated with its control strategy are potential early infection and level of immune response that may be high in one-day-old chicks due to maternally derived antibodies. Herein, two vaccination regimes were evaluated in commercial broilers kept under either field or laboratory conditions. Two different vaccine types and concentrations were used, and the results highlighted a significantly higher protection against early infection when a homologous vaccine of high antigenic mass was applied at 7 days of life. Shedding was significantly reduced in this regime.

**Abstract:**

Low pathogenic avian influenza virus is one of the major threats that has been affecting the poultry industry in the Middle East region for decades. Attempts to eradicate this disease have failed. Currently, there are commercial vaccines that are either imported or produced locally from recently circulating isolates of H9N2 in Egypt and Middle Eastern countries. This present work focused on comparing the effectiveness of two vaccines belonging to these categories in Egypt. Two commercial broiler flocks (Cobb-500 Broiler) with maternally derived immunity (MDA) against H9N2 virus were employed and placed under normal commercial field conditions or laboratory conditions. Immunity was evaluated on the basis of detectable humoral antibodies against influenza H9N2 virus, and challenge was conducted at 28 days of life using a recent wild H9N2 virus. The results showed that vaccination on the 7th day of life provided significantly higher immune response in both vaccine types, with significantly lower virus shedding compared to vaccination at day 1 of life, regardless of field or laboratory conditions. In addition, the vaccine produced from a recent local H9N2 isolate (MEFLUVAC-H9-16) provided a significantly higher humoral immune response under both field and laboratory conditions, as measured by serology and virus shedding (number of shedders and amount of shedding virus), being significantly lower following challenge on the 28th day of life, contrary to the imported H9 vaccine. In conclusion, use of H9N2 vaccine at 7 days of life provided a significantly higher protection than vaccination at day 1 of life in birds with MDA, suggesting vaccination regimes between 5–8-days of life for broiler chicks with MDA. Moreover, use of a vaccine prepared from a recently circulating H9N2 virus showed significantly higher protection and was more suitable for birds in the Middle East.

## 1. Introduction

The poultry industry has been suffering from several pathogens in Egypt during recent decades, including avian influenza viruses (AIV) that may be either highly pathogenic avian influenza (HPAIV; H5N1, H5N2, H5N8) or low-pathogenic avian influenza (LPAIV; H9N2) viruses, velogenic Newcastle disease virus (vNDV), infectious bronchitis virus (IBV; variant 1, variant 2, and classic wild virus), infectious bursal disease (either variant or virulent virus), multidrug-resistant bacteria (MDR; *Escherichia coli*, *Salmonella*, *Pasteurella*, etc.), in addition to coccidia species. All these pathogens have caused severe economic losses and have badly affected the veterinary care strategies [1,2,3,4,5,6,7,8,9,10]. Avian influenza (AI) is a contagious viral disease, belonging to the Orthomyxoviridae family, and is a segmented, single-stranded, negative sense RNA virus [11,12]. Avian influenza viruses (AIV) belong to type-A and are divided into subtypes on the basis of antigenic relationships of the surface glycoproteins (hemagglutinin and neuraminidase) into 18 hemagglutinins (HA) and 11 neuraminidases (NA), with variable combinations. These glycoproteins are considered the main antigenic components of the virus and act as immune modulators for pattern recognition receptors of the immune system. Structural variations within these pathogen surface glycoproteins were shown to affect the different host responses, whether they be avian, fish, or mammalian [13,14,15,16]. On the basis of the pathogenicity of the avian virus, it is further classified into two types known as a highly pathogenic avian influenza virus “H5/H7” (HPAIV) and a low pathogenic avian influenza virus (LPAIV) [12,17,18]. The H9N2 avian influenza virus is associated with one of the major viral problems affecting the poultry industry in Egypt since it was officially reported for the first time in 2011 until now [19]. Virus infection leads to high economic losses in both layers and for breeders due to a drop in egg production. Broilers may also show severe losses during co-infection with other pathogens, especially Infectious Bronchitis Virus (IBV),Newcastle disease virus (NDV), bacteria such as *E. coli* and *Mycoplasma*, or even live virus vaccines [5,9,20,21]. Recently several reports have highlighted the immunosuppressive effect associated with LPAI-H9N2 in poultry flocks either by altering the differentiation of lymphocytes or inflammatory cytokines or the depletion and apoptosis of some immune cells [22,23,24,25,26,27,28,29]. Moreover, some reports have discussed the effect of H9N2 on the alteration of blood biochemical and hematological parameters [30]. Regarding the potency and efficacy of the H9N2 inactivated vaccine in Egypt, there is a paucity of knowledge addressing this issue.

This work focuses on evaluating the value of applying the H9N2 vaccine at 1 day or 7 days of life on the developed immune response under both farm commercial conditions and laboratory standard conditions in broiler chicks with maternally derived immunity (MDA). Two commercially available vaccines were compared: one imported that is based on an old Middle East isolate (1998) and another produced in Egypt from a recent Middle East isolate (2016). Protection following challenge was assessed against live H9N2 virus in birds kept under standard laboratory conditions only.

## 2. Materials and Methods

### 2.1. Ethical Approval

Animal studies were approved by the Animal Welfare and Research Ethics Committee of Suez Canal University and all procedures were conducted strictly in accordance with the Guide for the Care and Use of Laboratory Animals. Every effort was made to minimize animal suffering.

### 2.2. Birds and Vaccines

A total of 108,120 one-day-old commercial Cobb-500 broilers from vaccinated broiler breeders with H9N2 vaccine was used in this experiment. All birds were given vaccines for infectious bursal disease (IBD), AIV-H5, and Newcastle disease (ND) as is routine for commercial broilers in Egypt. Two commercial inactivated H9N2 vaccines were used in this trial (vaccines A and B). Vaccine A (MEFLUVAC-H9ND-16) is produced by MEVAC Co. Egypt and prepared from the A/ck/Egypt/ME/543V/2016(H9N2) virus. Vaccine B (Gallimune 208 H9ND) is produced by Merial Incorporation, France, and was brought from a local agency in Egypt. Vaccine B is prepared from A/chicken/Iran/Av1221/1998. Both were used according to the manufacturer’s recommendations.

### 2.3. Experiment Design

#### 2.3.1. Laboratory Groups

A total of 120 one-day-old commercial Cobb-500 broiler chicks with maternally derived antibodies for H9N2 virus (MDA) were divided into 6 different experimental groups: group 1 was vaccinated with MEFLUVAC-H9ND-16 (vaccine A) and group-2 received an imported H9ND vaccine (vaccine B)—both were administered at day 1 of life. Group 3 took vaccine A and group 4 took vaccine B, but at the 7th day of life. Group 5 served as a positive control (non-vaccinated, challenged group) and group 6 served as a negative control (non-vaccinated, non-challenged group). All vaccines were provided as per the manufacturer’s recommendations, as shown in Table 1. Challenge was conducted in G1-5 (10 birds from each group) at the 28th day of life using a previously identified AIV-H9N2 virus [31].

#### 2.3.2. Field-monitored Groups

A total of 108,000 broiler chickens (Cobb-500) produced from vaccinated broiler breeders with H9N2 vaccine and showing maternally derived antibodies for H9N2 virus (MDA) were placed equally in 4 pens, each with 27,000 birds (G-7-10). Group 7 took a commercial local H9ND vaccine (A), while group 8 took a commercial imported H9ND vaccine (B); both products were administered at the 1st day of life by subcutaneous (S/C) injection using the manufacturer’s recommended dose. Group 9 took vaccine A while group 10 took vaccine B at the 7th day of life. Birds in groups 7–10 were brought from the same broiler breeder flock and the same hatchery and kept under commercial field conditions with proper biosecurity measures and took the same ratio and management standers.

### 2.4. Hemagglutination Inhibition (HI) Test

The HI test was used to monitor post-vaccination humoral immune response for each vaccine using avian influenza H9N2 antigens (one representative of the circulating virus in Egypt and another imported antigen). Chicken sera were examined for HA-specific antibodies against H9N2 virus by HI test according to the Office International des Epizooties (OIE) manual (OIE, 2015). Serial twofold serum dilutions in phosphate buffer saline (PBS) were subsequently mixed with equal volumes (25 μL) of the virus containing 4 hemagglutinating units (HAU), then 25 μL of washed chicken red blood cells were added. After we incubated the HI titers for 40 min at room temperature, we determined them as reciprocals of highest serum dilutions in which inhibition of hemagglutination was observed.

### 2.5. Challenge Virus

Vaccinated birds grown in isolated rooms were challenged at 4 weeks of age by intranasal inoculation of 6 log_10_ embryo infective dose_50_ (EID_50_) of the previously isolated wild type AI-H9N2 virus [31].

### 2.6. qRT-PCR for Virus Shedding

Tracheal swabs were collected from the challenged birds for detection of virus shedding by RT-PCR at 3 and 7 days post challenge, as per the OIE manual [32], using specific primers and probes, as previously described [31]. qRT-PCR titers were converted into log_10_ EID_50_/_mL_, as described previously [33]. Briefly, a triplicate of 6 10-fold dilutions of challenge AIV-H9N2 (AIV-H9N2; 10^6^ EID_50_/_mL_) were used to generate a standard curve using stock virus dilutions from 10^−1^ to 10^−6^. Since Ct is defined as the point at which the curve crosses the horizontal threshold line, we plotted virus log_10_ titers of a specimen against the Ct value, and the best fit line was constructed. The linear range of the assay ranged from 1 to 10^6^ EID_50_/_mL_, with a correlation coefficient of 0.99. System detection limit was 0.5 EID_50_/_mL_, as has been standardized and described previously [31]. AIV-H9N2 quantity in unknown samples were derived by plotting the Ct of an unknown against the standard curve and were expressed in log_10_ EID_50_/_mL_ equivalents.

### 2.7. Statistical Analysis

Where necessary, data were analyzed by Student’s *t*-test or by ANOVA followed by application of Duncan’s new multiple range test to determine the significance of differences between individual treatments and corresponding control [34].

## 3. Results

### 3.1. Different AIV-H9N2 Virus Hemagglutinin Segment Amino Acid Identity Degrees

The hemagglutination segment (HA) amino acids identity degree showed higher similarities with vaccine A seed, ranging from 93.8 to 98.8% with different isolated viruses from Middle Eastern countries while with vaccine B, the degree of similarity was lower as it ranged from 89.5 to 92.8%, as shown in Table 2.

### 3.2. Immune Response to Other Vaccines at 28th Day of Life

Birds in all groups showed immune responses for vaccinations with infectious bursal disease vaccine, avian Influenza-H5 vaccine, and Newcastle disease vaccines. No significant differences were found among the different groups, as is shown in Table 3.

### 3.3. Immune Response in Groups Kept under Laboratory Conditions

HI assay using antigen A (representing recently circulating H9N2 virus in the Middle East), not H9N2 virus challenge, revealed that birds in groups 3–4 (vaccinated at 7 days of life) had significantly higher (*p* ≤ 0.05) immune responses at 28 and 35 days of life, compared to groups 1 and 2 that were vaccinated at day 1 of life. Moreover, group 1 (vaccine A) showed a significantly higher immune response (*p* ≤ 0.05) at 21, 28, and 35 days of life compared to birds in group 2 (vaccine B). At 7 and 14 days of life, birds in group 1 showed a higher but not significantly different (*p* value ≥ 0.05) immune response compared to group 2, as shown in Figure 1 and Table 3. Birds in group 3 (vaccine A at D-7) showed a significantly higher immune response (*p* ≤ 0.05) at 28 and 35 days of life in comparison with birds in group 4 (vaccine B). Yet, at 7, 14, and 21 days of life, birds in group 3 showed a higher but non-significant difference in the immune response (*p* ≥ 0.05) compared to group 4, as shown in Figure 1 and Table 3. Birds in groups 5 and 6 showed non-detectable (nt) HI titer at 21 and 28 days of life, which reflected complete weaning from the maternally derived antibodies at 21 days of age. Birds in group 6 showed an undetectable immune response at 35 days of life, which ensured a negative control condition. Birds in group 6 developed seroconversion at 35 days of life (7 days post-challenge), which reflected the positive effect of the challenge virus. Performing HI assay using antigen B (representing the imported vaccine, not the recently circulating virus in the Middle East) without challenge showed that birds in groups 3–4 (vaccinated at the 7th day of life) developed a significantly higher (*p* ≤ 0.05) immune response at 28 and 35 days of life versus groups 1-2 that were vaccinated at day 1 of life. There was no significant difference between the HI titers of the different groups vaccinated at the same age. Birds in group 5 showed significantly lower HI titers using antigen B as compared with antigen A at 7 days post-challenge, as shown in Table 4.

### 3.4. Protection Following Challenge with a Wild Type H9N2 at the 28th Day of Life

#### 3.4.1. Seroconversion 7-DPC with Recent Middle Eastern H9N2

At 7-DPC (days post-challenge), birds in group 5 (non-vaccinated, challenged) showed seroconversion with an average Geometric mean titer (GMT) HI titer of 8.3 ± 0.65 using antigen representing the recently circulating H9N2 virus in Egypt. At the same time, the titer was 7.4 ± 0.82 using the standard imported H9N2 antigen. Birds in group 2 and group 4 showed a significant increase in immune response (HI titer) at 7-DPC in comparison with birds of the same group and age, but unchallenged. Birds in group 1 and group 3 at 7-DPC demonstrated a declined immune response compared to those unchallenged in the same group and age, as shown in Table 5.

#### 3.4.2. Virus Shedding Following Challenge with H9N2 at 28 Days of Life

Birds in groups 3 and 4 (vaccinated at 7 days of life) showed a significantly lower number of shedders at 3-DPC in comparison with groups 1 and 2 (vaccinated at day 1 of life). Of groups vaccinated on the first day of life, group 1 (vaccine A) showed a significant reduction in virus shedding; number of shedders and amount of shed virus at 3/7-DPC compared to group 2 (vaccine B) (Table 6). Birds in group 3 (vaccine A at D-7) showed a significant reduction in virus shedding in terms of number of shedders and amount of shed virus at 3/7-DPC compared to group 2 (vaccine B), as shown in Table 6. Birds in group 2 (vaccine B) showed a reduction in the number of shedders compared to birds in group 5 (non-vaccinated, challenged), while there was no significant difference in the amount of virus shedding via the cloacal route at 3-DPC between birds in group 2 (vaccine B at day 1 of life) and group 5 (non-vaccinated, challenged). Moreover, birds in both groups (group 2 and group 5) showed the same virus shedding amount via tracheal route at 7-DPC, while birds in group 6 showed non-detectable (nd) virus shedding at 3- and 7-DPC in both tracheal and cloacal swaps.

### 3.5. Immune Response in Groups Kept under Field Condition

Birds in groups 9–10 (vaccinated at 7 days of life) showed a significantly higher immune response at 28 and 35 days of life in comparison with groups 7 and 8 that were vaccinated at day 1 of life. In groups vaccinated at the first day of life, group 7 (vaccine A) showed a significantly higher immune response (*p* ≤ 0.05) at 28 and 35 days of life compared to birds in group 8 (vaccine B), while at 7, 14, and 21 days of life, birds in this group showed a non-significant increase (*p* ≥ 0.05) in the immune response versus group 8 (Figure 2). Birds in group 9 (vaccine A at 7 days of life) showed a significantly higher immune response (*p* ≤ 0.05) at 28 and 35 days of life in comparison with birds in group 10 (vaccine B), while at 14 and 21 days of life, birds in group 9 showed a non-significant increase (*p* ≥ 0.05) in the immune response compared to group 10 (Figure 2).

## 4. Discussion

Broiler chicks in Egypt and the Middle East almost always arise from broiler-breeders’ flocks vaccinated against H9N2. Accordingly, the vast majority of one-day-old broiler chicks produced in the region carry MDA against H9N2, which compromises the use of the inactivated H9N2 vaccines [29,31]. However, the immune response to other applied vaccines (IBD, NDV, H5) among all bird groups showed detectable antibody levels, with no significant differences after receiving one of the two vaccines employed either at day 1 of life or 7 days of life, or among non-vaccinated controls, which agrees with the previous reports of Kilany et al., Sultan et al., and Khalil et al., who claimed that applying vaccines does not interfere with the immune response of other different vaccines [30,35,36].

Commercial broilers with MDA against AIV-H9N2 face difficulties in developing an immune response following vaccination with inactivated H9N2 vaccines at day 1 of life due to interference. However, the most common regime for H9N2 vaccination in Egypt is to administer a single-dose between days 1–5 of life in broiler sectors, resulting in frequent vaccine failure. Results from the present study show that commercial broilers with MDA against H9N2 develop significantly higher immune response when applied at 7 days of life rather than day 1 of life, regardless the vaccine type used (either A or B). This result explains in part the repeated H9N2 vaccination failure in commercial boiler flocks with MDA (vaccinated at day 1 of life). Interference of the vaccine with the high titer of MDA against H9N2 (average = 6–8 log_2_) leads to this failure, as previously reported [37,38,39]. Under standard laboratory conditions, birds receiving vaccine A developed significantly higher immune response compared to vaccine B when administered at the same age (day 1 of life and 7 days of life). This may be referred to the amount of antigen in each vaccine, with vaccine A containing a higher amount of antigen (350 HAU unit/dose), while vaccine B contained around 200 HAU unit/dose [38]. As previously reported by Kilany et al. (2016), increasing the dose from 200 to 250 or 350 HAU can improve the immune response and protection against H9N2 [6].

Virus shedding is an important factor in the epidemiology of avian influenza; the lower the amount of virus shedding (amount of virus shedding and number of shedders’ birds), the better we can control avian influenza. Results from the current study showed that use of the inactivated H9N2 vaccine at 7 days of life significantly reduced the virus shedding at 3- and 7-DPC compared to the inoculation at day 1 of life. Birds receiving vaccine A at day 1 of life significantly showed lower virus shedding in amount and number of shedders compared to those receiving vaccine B at the same age. However, birds receiving vaccine B at day 1 of life showed a significant reduction in virus shedding compared to the non-vaccinated challenge group, with tracheal shedding at 3-DPC and cloacal shedding at 7-DPC. No significant virus shedding was observed in the tracheal and cloacal swabs at 7-DPC between group 5 (non-vaccinated challenged) and group 2 (vaccine B at day 1 of life), which may have been due to the lower antigenic mass in the vaccine dose and higher level of MDA at day 1 of life. This negatively impacts the development of immune response, in agreement with the previous reports by Kilany et al. (2016), who demonstrated that lower antigenic masses (less than 128–200 HAU/dose) do not significantly reduce virus shedding compared to non-vaccinated challenged birds [40].

Applying vaccine-A at 7 days of life showed a significant reduction in virus shedding at 3- and 7-DPC compared to administering it at day 1 of life. This is contrary to the findings from vaccine B at 1 or 7 days of life, which may be explained by the higher antigenic mass in vaccine A along with the decline of MDA at 7 days of life, consent with Sun et al. (2012) and Khalil et al. (2015), who reported a significantly higher protection in chickens with H9N2 vaccines containing a higher antigenic mass than 250 HAU/dose. Moreover, Elfeil et al. (2019) reported a significant reduction in virus shedding in turkey following vaccination with H9N2 vaccine (350 HAU/dose) [36,40,41].

Commercial farm conditions always differ from laboratory conditions. In this study, birds kept under farm conditions and vaccinated at 7 days of life showed a significantly higher immune response than those vaccinated at day 1 of life. This matches with the results of laboratory groups and confirms the observation that vaccination at 7 days of life provides a significantly higher immune response and expected protection [36,42]. In addition, groups kept under farm commercial conditions of mass production (25,000 birds/pen) showed 1–3 log_2_ lower HI titers than those kept under laboratory conditions, indicating that there is around 7–25% difference in expected immune response between farm and laboratory conditions. This finding highlights the need to perform more research trials under commercial farm conditions, including virus challenge under Biosafety level-3 (BSL-3) isolators, bearing in mind that the negative pressure inside the isolators may affect virus spreading, as mentioned previously in the case of AIV-H9N2 in turkey poults, as well as in Newcastle disease virus and infectious bronchitis virus in chicken [40,41,42,43,44].

## 5. Conclusions

Application of AIV-H9N2 inactivated vaccine at 7 days of life provides a significantly higher protection on the basis of antibody level and reduction of virus shedding, number of shedders, and amount of virus shed per bird versus vaccines given at day 1 of life. Use of a homologous vaccine with high antigenic mass could also help in the reduction virus shedding and provide a significantly higher immunity and protection. Application of such a regime could help the control strategies of AIV-H9N2 in commercial broiler flocks in endemic areas and reduce the epidemiological load of AIV-H9N2 virus in the environment.

## Figures and Tables

**Figure 1 animals-10-01875-f001:**
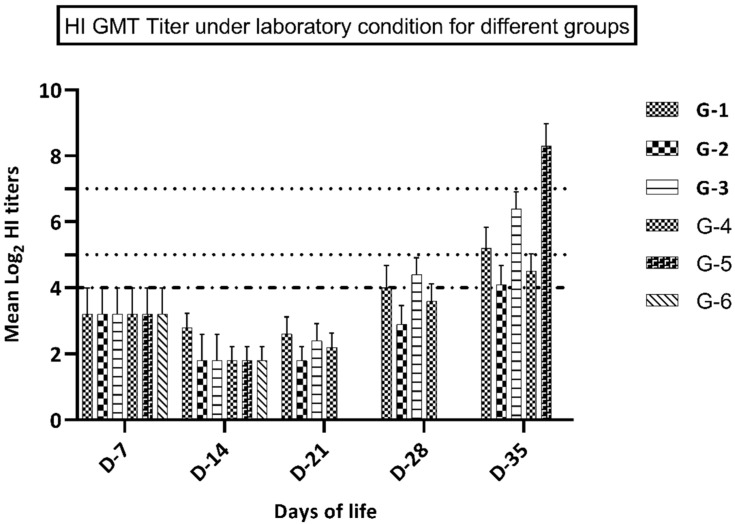
GMT of HI titers in groups 1-6 (kept under laboratory conditions).

**Figure 2 animals-10-01875-f002:**
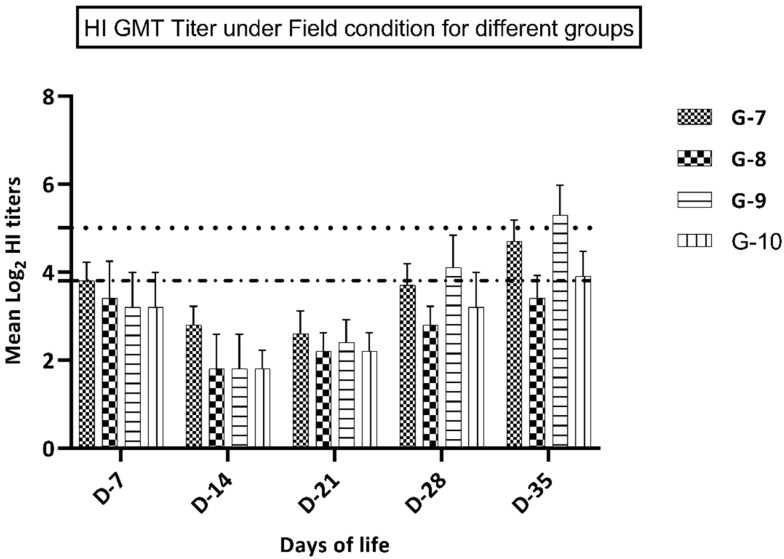
GMT HI titer in groups 7–10 (kept under field conditions).

**Table 1 animals-10-01875-t001:** Experimental design for different groups (G-1-10).

	Group No.	Bird No.	Vaccine Regime			Challenge at 28 Day of Age	Assessment of Protection
			Vaccine Type	Age/Days	Dose/mL		
Experiment 1 lab experiment	G-1	20	A	1	0.3	+++	Follow up of immune response of vaccinated birds at weeks post vaccination to H9, H5, ND, IBD, and IB vaccines using HI and ELISA tests. Viral shedding detected by real-time PCR.
	G-2	20	B	1	0.3	+++	
	G-3	20	A	7	0.3	+++	
	G-4	20	B	7	0.3	+++	
	G-5	20	-	------	----	+++	
	G-6	20	-	------	---	-----	
Experiment 2 field group	G-7	27K	A	1	0.3	-	Follow up of immune response of vaccinated birds on weekly basis. Measure of performance parameters and field exposure. Natural exposure monitoring by RT-PCR.
	G-8	27K	B	1	0.3	-	
	G-9	27K	A	6	0.3	-	
	G-10	27K	B	6	0.3	-	

**Table 2 animals-10-01875-t002:** Different avian influenza (AIV)-H9N2 virus hemagglutinin segment amino acid identity degrees.

Items	Alg. 2017	Egypt 2018	Iraq 2017	KSA 2018	Leb. 2017	Libya 2015	Mor. 2018	Pak. 2018	Tun. 2015	UEA 2017	Vaccine A	Vaccine B
**Alg./2017**		94.59	94.78	96.39	94.59	96.39	99.28	91.88	91.76	98.10	93.68	92.24
**Egypt/2018**	94.59		94.79	93.68	94.64	95.54	94.64	92.54	94.58	96.46	98.86	91.79
**Iraq/2017**	94.78	94.79		94.34	93.32	93.93	93.21	94.46	93.11	93.04	94.43	90.71
**KSA/2018**	96.39	93.68	91.34		94.22	95.49	96.57	90.43	92.31	96.39	94.78	91.88
**Leb./2017**	94.59	94.64	92.32	94.22		96.79	94.82	90.89	93.41	94.64	94.75	92.86
**Libya/2015**	96.39	95.54	93.93	95.49	96.79		96.25	92.86	95.33	96.07	94.64	93.93
**Mor./2018**	99.28	94.64	93.21	96.57	94.82	96.25		91.96	92.31	99.82	94.75	92.14
**Pak./2018**	91.88	90.54	94.46	90.43	90.89	92.86	91.96		89.84	91.79	94.18	89.11
**Tun./2015**	91.76	92.58	90.11	92.31	93.41	95.33	92.31	89.84		92.31	95.48	89.56
**UEA/2017**	98.10	94.46	93.04	96.39	94.64	96.07	99.82	91.79	92.31		96.57	91.96
**Vaccine A**	94.68	98.86	94.43	94.78	93.75	94.64	94.75	94.18	95.48	96.57		90.61
**Vaccine B**	92.24	91.79	90.71	91.88	92.86	93.93	92.14	89.11	89.56	91.96	91.61	

**Alg.:** Algeria; **Leb.:** Lebanon; **Mor.:** Morocco; **Pak.:** Pakitsan.

**Table 3 animals-10-01875-t003:** Serological response of broiler chickens to infectious bursal disease virus (IBDV), Newcastle disease (ND), and avian influenza-H5N1 (AI-H5) vaccines at 28 days of age.

Group No.	Bird No.	Vaccine Regime			ELISA Mean Titers	HI Titer Log_2_	
		Vaccine Type	Age/days	Dose/mL	IBD	AI-H5	ND
					28 Days	28 Days	28 Days
G-1	20	A	1	0.3	17,553 ± 1105	3.7 ± 0.51	4.8 ± 0.72
G-2	20	B	1	0.3	17,703 ± 1120	3.8 ± 0.61	4.9 ± 0.61
G-3	20	A	7	0.3	17,612 ± 1090	3.6 ± 0.42	4.7 ± 0.66
G-4	20	B	7	0.3	16,217 ± 1220	3.6 ± 0.52	4.8 ± 0.62
G-5	20	-	-	-	17,533 ± 1140	3.7 ± 0.65	4.9 ± 0.56
G-6	20	-	-	-	17,533 ± 1170	3.7 ± 0.72	4.8 ± 0.81

ELISA = enzyme-linked immunosorbent assay; HI = hemagglutination inhibition test; IBDV = infectious bursal disease virus; ND = Newcastle disease virus; AI-H5 = avian influenza-H5N1.

**Table 4 animals-10-01875-t004:** Hemagglutination inhibition (HI) test results in bird groups (G-1-6) using two-antigens.

Days of Life	Antigen Type	Experimental Groups					
		G-1	G-2	G-3	G-4	G-5	G-6
D-7	Antigen A	3.2 ± 0.79	3.2 ± 0.79	3.2 ± 0.79	3.2 ± 0.79	3.2 ± 0.79	3.2 ± 0.79
	Antigen B	2.6 ± 0.69	2.6 ± 0.69	2.6 ± 0.69	2.6 ± 0.69	2.6 ± 0.69	2.6 ± 0.69
D-14	Antigen A	2.8 ± 0.54	1.8 ± 0.42	1.8 ± 0.49	1.8 ± 0.56	1.8 ± 0.48	1.8 ± 0.32
	Antigen B	2.1 ± 0.44	2.2 ± 0.54	1.6 ± 0.74	1.6 ± 0.64	1.6 ± 0.74	1.6 ± 0.64
D-21	Antigen A	2.6 ± 0.55	1.8 ± 0.67	2.4 ± 0.52	2.2 ± 0.42	nd	nd
	Antigen B	2.2 ± 0.54	2.3 ± 0.64	2.1 ± 0.54	2.1 ± 0.54	nd	nd
D-28	Antigen A	4.1 ± 0.71	2.9 ± 0.59	4.4 ± 0.57	3.6 ± 0.46	nd	nd
	Antigen B	3.1 ± 0.44	3.1 ± 0.54	3.4 ± 0.64	3.2 ± 0.64	nd	nd
D-35	Antigen A	5.4 ± 0.61	4.0 ± 0.57	6.4 ± 0.52	4.5 ± 0.67	8.3 ± 0.65	nd
	Antigen B	4.2 ± 0.54	4.1 ± 0.48	5.1 ± 0.45	4.9 ± 0.64	7.4 ± 0.82	nd

Antigen A: antigen prepared from recently circulating H9N2 virus similar to vaccine A seed virus; antigen B: imported antigen representing vaccine B seed virus; D-7: 7 days of life; D-14: 14 days of life; D-21: 21 days of life; D-28: 28 days of life; D-35: 35 days of life; nd: non-detectable level; G-1: vaccine A at D-1; G-2: vaccine B at D-1; G-3: vaccine A at D-7; G-4: vaccine B at D-7; G-5: non-vaccinated, challenged (positive control); G-6: non-vaccinated, non-challenged (negative control).

**Table 5 animals-10-01875-t005:** The seroconversion at 7-DPC (days post-challenge) with recent Middle East H9N2 virus with two types of AIV-H9 antigen.

Group No.	Bird No.	Vaccine Regime			GMT Log_2_ HI Titer (*n* = 10)	
		Vaccine Type	Age/days	Dose/mL	7 Days Post-Challenge	
					Local Antigen	Imported Antigen
1	10	A	1	0.3	4.1 ± 1.2 *	3.2 ± 1.21 *
2	10	B	1	0.3	7.0 ± 2.57 *	5.9 ± 2.57 *
3	10	A	7	0.3	5.1 ± 0.92 *	4.8 ± 0.89 *
4	10	B	7	0.3	6.5 ± 1.67 *	5.5 ± 1.87 *
5	10	-	-	-	8.3 ± 0.65 *	7.4 ± 0.82 *
6	10	-	-	-	-	-

*: mean there is a significant difference between means when *p* > 0.05.

**Table 6 animals-10-01875-t006:** Virus shedding at 3- and 7-days post-challenge.

Group No.	Vaccinal Regime		Assessment of Protection							
	Vaccine Type	Vaccine Age	3-DPC				7-DPC			
			Tracheal Swabs		Cloacal Swabs		Tracheal Swabs		Cloacal Swabs	
			No./EID50	%	No./EID50	%	No./EID50	%	No./EID50	%
G-1	A	1	4/10 (2.1 ± 0.9) ^b^	40%	3/10 (1.5 ± 0.3) ^c^	30%	3/10 (1.8 ± 0.7) ^b^	30%	3/10 (1.8 ± 0.8) ^b^	30%
G-2	B	1	6/10 (2.8 ± 0.8) ^c^	60%	4/10 (2.1 ± 0.4) ^c^	40%	3/10 (2.3 ± 1.1) ^c^	30%	4/10 (2.5 ± 0.8) ^c^	40%
G-3	A	7	2/10 (1.6 ± 0.6) ^a^	20%	2/10 (1.1 ± 0.3) ^a^	20%	1/10 (1.3 ± 0.0) ^a^	10%	2/10 (1.8 ± 0.5) ^a^	20%
G-4	B	7	4/10 (2.4 ± 0.7) ^c^	40%	3/10 (1.9 ± 0.3) ^c^	30%	2/10 (1.9 ± 0.7) ^c^	20%	3/10 (2.1 ± 0.6) ^c^	30%
G-5	-	-	10/10 (3.1 ± 0.9) ^d^	100%	6/10 (2.1 ± 0.3) ^c^	60%	7/10 (2.3 ± 1.9) ^c^	70%	10/10 (3.1 ± 1.8) ^d^	100%
G-6	-	-	nd	-	nd	-	nd	-	nd	-

^abcd^ means different superscripts differ significantly (*p* ≤ 0.05); DPC: days post-challenge; EID_50_: egg infectious dose_50_; nd: non-detectable level; G-1: vaccine A at day 1; G-2: vaccine B at day 1; G-3: vaccine A at day 7; G-4: vaccine B at day 7; G-5: non-vaccinated, challenged (+ control); G-6: non-vaccinated, non-challenged (control).

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
