# Peer review of "Comparison of the Effectiveness of Two Different Vaccination Regimes for Avian Influenza H9N2 in Broiler Chicken"

_animals, 2020, doi:10.3390/ani10101875_

Round 1
Reviewer 1 Report
In this study, two vaccination regimes were evaluated in commercial broiler in both field farm condition and laboratory condition using two different concentration of vaccines, the results highlighted that significant higher protection against early infection could obtained by using homologous vaccine with high antigenic mass applied at 7-day of life. The paper might be helpful to apply vaccines controlling avian influenza caused by H9N2 virus in Egypt. The paper might be accepted after a major revision.
Major comments:
- Besides vaccine-A contain higher amount of antigen (350HAU unit/ dose) while vaccine-B contain around 200 HAU unite/dose (lines 236-237), the genetic lineages based on HA gene of these two vaccine strains should be presented since the efficacy of different H9N2 vaccines might be influenced by the HA gene variation between the vaccine strain and the circulating strain. For this concern, in “2.5. Challenge virus” (lines 129-131), the HA similarity between the challenge strain and two vaccine strains should be detailed.
- The manuscript should be polished by a native English speaker.
- Lines 132-135, in “2.6. qRT-PCR for virus shedding”, authors described “Tracheal swabs were collected from the challenged birds for detection virus shedding by RT-PCR at 3- and 7-days,”, however, in “3.3.2. Virus shedding following challenge with H9N2 at 28-day of life”, “EID50”was employed to indicate the level of virus shedding, authors should check and confirm the detection method.
- Lines 177-181, “Birds in G-2 and G-4 showed significant increase in immune response (HI titer) at 7-DPC in comparison to birds from same group at same age but not challenged’ while birds on G-1 and G-3 at 7-DPC showed decline in the immune response in comparison to birds from the same group at same age but not-challenged as shown in table 4.”, HI titer at 7-DPC in birds from same group at same age but not challenged’ were not presented in table 4 or in context, authors should confirm and supplement them.
Minor comments:
- Lines 57-58, “Escherichia Coli, Salmonella, Pasteurella” should be italic. Also in line 74, “E. Coli, Mycoplasma”.
- Line 77, “repots” should be“reports”.
- Line 80, “describes” should be“described”.
- Line 99, “imported H9ND vaccine” should be “imported H9ND vaccine (Vaccine-B)” in accordance with “MEFLUVAC-H9ND-16 (Vaccine-A)”.
- In table 2, “IBDV” should be“IBD”?
- Page 6, the title of table 5 was lost.
- Page 6, 7, the antibody titer significance between different groups should be indicated in Fig.1 and Fig.2, and also noted in the figure legends.
Author Response
Dear Dr
Greeting
On behalf of the author, we appreciate your valuable comments, which support us to improve the quality of our manuscript
We hope our manuscript takes your consideration
Reviwer-1 response
Major comments:
- Besides vaccine-A contain higher amount of antigen (350HAU unit/ dose) while vaccine-B contain around 200 HAU unite/dose (lines 236-237), the genetic lineages based on HA gene of these two vaccine strains should be presented since the efficacy of different H9N2 vaccines might be influenced by the HA gene variation between the vaccine strain and the circulating strain. For this concern, in “2.5. Challenge virus” (lines 129-131), the HA similarity between the challenge strain and two vaccine strains should be detailed.
Added: line 135-149 - The manuscript should be polished by a native English speaker.
done - Lines 132-135, in “2.6. qRT-PCR for virus shedding”, authors described “Tracheal swabs were collected from the challenged birds for detection virus shedding by RT-PCR at 3- and 7-days,”, however, in “3.3.2. Virus shedding following challenge with H9N2 at 28-day of life”,
3- and 7-days post challenge “added”
“EID50”was employed to indicate the level of virus shedding, authors should check and confirm the detection method.
Detection method by qRT_PCR and converted to EID50 based on previously prepared standard curve on the same PCR machine with the same challenge virus and used molecular materials in our lab. “line 121-128” - Lines 177-181, “Birds in G-2 and G-4 showed significant increase in immune response (HI titer) at 7-DPC in comparison to birds from same group at same age but not challenged’ while birds on G-1 and G-3 at 7-DPC showed decline in the immune response in comparison to birds from the same group at same age but not-challenged as shown in table 4.”, HI titer at 7-DPC in birds from same group at same age but not challenged’ were not presented in table 4 or in context, authors should confirm and supplement them.
Pointed in table-4 with two type of antigen, one represent challenge virus (local antigen) and one represented the imported vaccine (imported antigen)
Minor comments:
- Lines 57-58, “Escherichia Coli, Salmonella, Pasteurella” should be italic. Also in line 74, “E. Coli, Mycoplasma”.
Changed to italic - Line 77, “repots” should be“reports”.
Changed - Line 80, “describes” should be“described”.
changed - Line 99, “imported H9ND vaccine” should be “imported H9ND vaccine (Vaccine-B)” in accordance with “MEFLUVAC-H9ND-16 (Vaccine-A)”.
added - In table 2, “IBDV” should be“IBD”?
changed - Page 6, the title of table 5 was lost.
added

Reviewer 2 Report
The authors in this paper provide the experimental data on the Comparison of the effectiveness of two different vaccination regimes for avian influenza H9N2 in broiler Chicken. Results clearly showed using a vaccine prepared from the recent H9N2 virus isolates showed significantly higher protection and more suite the Middle East recent circulating H9N2 virus, especially when used at the 7th day of life which provides significant protection than using at the first day of life in birds with maternal-derived immunity.
The science and technical execution of the study is of good quality. The study is solid and the data, in general, support the conclusions. The theory, logic, and experimental design are easy to follow and in general, make sense.
General comments: 1- the authors used very long sentences in many areas of the manuscript, therefore, rephrase in necessary. 2- Major English language revision of the whole manuscript is mandatory. 3- The authors used different abbreviations for the same terms, like the word (group and non-detectable,...etc). 4- in Abstract: no need for the second sentence, delete it. Also, at line 33, delete (and both vaccine.........countries). 5- Line 38-45: rephrase and check language and use short simple sentences. 6- The word live in all abstract, change it to life. 7- Day of administrations: in manuscript, use the correct words like the 7th or day 7 of life. 8- Capitalization at the beginning of the sentences and in the middle of sentences, recheck. 9- The name of the bacteria must be italic. 10- Make separate sections of chickens and vaccines before section 2.2. 11- Change section 2.2. name to experimental design. 12- Section 2.2. is repeated on page 3, lines 95 and 115. 13- Line 105: change filed to Field. Line 113: standers to standards. 14- Where is the statistical analysis data, make a section for it. 15- Line 138: Show in the methodology that birds got vaccines for IBD, Avian influenza, and ND. 16- Line 148: delete (which vaccinated at day one of life), it is repetition. 17- Add significant marks to tables and figures. 18- Line 221" the word (Immune response) is a repetition, delete it. 19- Lines 223-232: very long sentence, rephrase, and use simple sentences. Overall, I believe the improved version of the manuscript will be of interest to the field of broiler chickens vaccination in veterinary sciences. Therefore, it should be recommended for publication in Animals after moderate revision.Author Response
Dear Dr
Greeting
On behalf of the author, we appreciate your valuable comments, which support us to improve the quality of our manuscript
We hope our manuscript takes your consideration
Reviwer-2 response
General comments:
- the authors used very long sentences in many areas of the manuscript, therefore, rephrase in necessary.
rephased - Major English language revision of the whole manuscript is mandatory.
Applied
- The authors used different abbreviations for the same terms, like the word (group and non-detectable,...etc).
uniformed - in Abstract: no need for the second sentence, delete it. Also, at line 33, delete (and both vaccine.........countries).
Done
- Line 38-45: rephrase and check language and use short simple sentences.
Done
- The word live in all abstract, change it to life.
Changed
- Day of administrations: in manuscript, use the correct words like the 7th or day 7 of life.
Used 7th day
- Capitalization at the beginning of the sentences and in the middle of sentences, recheck.
Checked and corrected
- The name of the bacteria must be italic.
Changed to italic
- Make separate sections of chickens and vaccines before section 2.2.
Added : line 77-82
- Change section 2.2. name to experimental design.
Changed - Section 2.2. is repeated on page 3, lines 95 and 115.
Deleted
- Line 105: change filed to Field. Line 113: standers to standards.
Changed
- Where is the statistical analysis data, make a section for it.
Added, line 120-123
- Line 138: Show in the methodology that birds got vaccines for IBD, Avian influenza, and ND.
Added: line 77-78
- Line 148: delete (which vaccinated at day one of life), it is repetition.
Deleted
- Add significant marks to tables and figures.
Added
- Line 221" the word (Immune response) is a repetition, delete it.
Deleted
- Lines 223-232: very long sentence, rephrase, and use simple sentences.
Rephrased

Round 2
Reviewer 1 Report
Line 151, "Vaccine-A seed ranged from 93.75-98.75......" should be "Vaccine-A seed ranged from 93.8-98.8%......" in line with "89.5-92.8 % as shown in......" in line 152.